# Stability of Hydroxy-α-Sanshool in Medium-Chain Triglyceride Oil and Corresponding Oil/Water Emulsions

**DOI:** 10.3390/foods12193589

**Published:** 2023-09-27

**Authors:** Takahiko Mitani, Yasuko Yawata, Nami Yamamoto, Mitsunori Nishide, Hidefumi Sakamoto, Shin-ichi Kayano

**Affiliations:** 1Center of Regional Revitalization, Research Center for Food and Agriculture, Wakayama University, Wakayama 640-8510, Japan; 2Faculty of Education, Wakayama University, Wakayama 640-8510, Japan; namiyama@wakayama-u.ac.jp; 3Division of Food and Nutrition, Wakayama Shin-Ai Women’s Junior College, Wakayama 640-0341, Japan; nishide37m@gmail.com; 4Faculty of Systems Engineering, Wakayama University, Wakayama 640-8510, Japan; skmt@wakayama-u.ac.jp; 5Department of Nutrition, Faculty of Health Science, Kio University, Nara 635-0832, Japan; s.kayano@kio.ac.jp

**Keywords:** *Zanthoxylum pipritum*, stability, sanshools, hydroxy-α-sanshool, α-tocopherol, *o*/*w* emulsion

## Abstract

The pungent component of sansho (Japanese pepper, *Zanthoxylum pipritum*) is sanshool, which is easily oxidized and decomposed. We have previously reported several sanshool stabilizers, such as α-tocopherol (α-Toc). Sansho pericarp powder treated with middle-chain triglycerides (MCTs) can be used to obtain extracts containing hydroxy-α-sanshool (HαS). Although HαS is stabilized when α-Toc is added to the MCT extracts, the loss of HαS is accelerated when it is mixed with a powder such as lactose. The separation of α-Toc from sanshools was thought to inevitably lead to their oxidation. Therefore, using sansho pericarp MCT extracts with or without α-Toc, oil/water (o/w) emulsions were prepared by adding a surfactant, glycerin, and water to these extracts. In both emulsions, HαS was stable in accelerated tests at 50 °C. However, when lactose powder was added to the emulsions and an accelerated test was performed, HαS in the emulsion containing α-Toc was stable, but HαS in the emulsion without α-Toc was unstable. These results highlight the importance of maintaining the close proximity of HαS and α-Toc in the emulsion. The stabilization of sanshools using emulsion technology can facilitate the production of various processed beverages, foods, cosmetics, and pharmaceuticals containing Japanese pepper.

## 1. Introduction

The fruits of Japanese pepper (sansho, *Zanthoxylum piperitum*) have been used since ancient times as a traditional remedy in Japan (Kampo medicine and herbal medicine). Sansho has a long history as a spice, and its special refreshing aroma and taste have attracted Japanese people. In addition to being used in Japanese cuisine, it is also used as a new flavor in French cuisine and chocolate, and its adoption is expected to increase in the future. In Japan, in preparations using sansho as a raw material, its fruits and pericarps are pulverized and/or extracted. However, while dried sansho fruits and pericarps have relatively stable taste, aroma, color, and other substances, powdered sansho pericarps rapidly lose their flavor, necessitating methods for preventing decomposition and stabilizing the spice components of sansho.

The pungent component of these fruits is *N*-alkylamide, which is called sanshool, and 13 sanshool isomers have been identified in the genus *Zanthoxylum* [1,2]. Sansho is a pungent spice with a tingling sensation and numbness, which are related to the type of sanshool. Hydroxy-α-sanshool (HαS) is the predominant sanshool in sansho. HαS is also the major sanshool in Chinese pepper (huajiao, Sichuan peppercorn, *Zanthoxylum bungeanum*) [3,4,5], and in fully ripe Chinese pepper, which is used most frequently, in addition to HαS, its structural isomer, hydroxy-β-sanshool (HβS), is also abundant. Huajiao is mainly produced in China, with an annual production of 300,000–400,000 tons. In addition to the use of dried pericarps as raw materials in herbal medicines, huajiao is also used as a seasoning and directly added to dishes, vegetable oil, and chili oil. The pungent component contains other *N*-alkylamides such as bungenols in addition to sanshool. Because these compounds are also unstable, they are utilized without crushing the pericarp. However, when seeking further processed products, a process of grinding and extracting the pepper is also required, and we believe that stabilization of the spice component is unavoidable.

A recent study reported a screening method for sanshool-stabilizing compounds and stabilizing substances [6]. In addition, the effect of pH on the stabilization of sanshools has been reported [6,7]. In brief, because sanshools contain unsaturated fatty acids in their structure, the addition of antioxidants is beneficial for preventing the oxidation and degradation of sanshools. Alpha-tocopherol (α-Toc), a gallic acid derivative; quercetin; and other antioxidants from the genus *Zanthoxylum* showed excellent sanshool-stabilizing action [6]. Among these, α-Toc was found to be useful because it is relatively abundant in sansho pericarps [8]. If the extracts of sansho pericarps (stock solution) are used to make food or drinks, it is necessary to dilute the stock solution with a powder or liquid. However, diluting the stock solution in this manner renders sanshools extremely unstable. This may occur because diluting the extract (stock solution) with a powder or liquid reduces the concentration of α-Toc, a stabilizing substance in sanshools, making the sanshools more susceptible to oxidation. To stabilize sanshools, it is necessary for a stable substance to always coexist closely with the sanshools and maintain its concentration.

The use of microencapsulation technology in food has recently gained popularity [9,10]. Attempts have been made to microencapsulate useful components of food, such as aromatic components, pigments, nutrients like unsaturated fatty acids, vitamins, antioxidants, etc., to stabilize them [9,10]. Conversely, microcapsules can also be used to trap unpleasant tastes and smells. Microencapsulation techniques are classified into several categories. Among the several types of microencapsulation technologies available at present, we used emulsion technology to attempt to stabilize sanshools. Sanshools can be extracted from sansho pericarps using vegetable oil, and α-Toc is highly soluble in vegetable oil. Therefore, we tried preparing o/w emulsions by vigorously mixing a sansho medium-chain triglyceride (MCT) ± α-Toc extracts with surfactants, glycerin, and water. The emulsions were subjected to various temperatures and pH conditions, and the amount of sanshool in the emulsion was tracked. In this experiment, the independent variables were (1) the type of solvent used to extract sansho pericarps when edible oil (MCTs) was used as the solvent in the extracts, (2) the effects of adding α-Toc when sansho MCT extracts were powdered, (3) the effects of adding α-Toc and temperature, (4) the type of surfactant used in emulsions of sansho MCT extract, (5) the effect of α-Toc in each emulsion, and (6) the effect of α-Toc in the emulsion when the emulsion was powdered, and all dependent variables were the HαS content.

This emulsion technology revealed that by keeping sanshools and the stabilizing substances in constant proximity, the stabilization of sanshools can be dramatically improved. In addition, since this is an o/w emulsion, it can be mixed with powder or dispersed in an aqueous solution, and it is expected to be used not only in various foods but also in pharmaceuticals and cosmetics.

## 2. Materials and Methods

### 2.1. Materials and Chemicals

Sansho fruits (variety name: Budo Sansho) were harvested from Kainan, Wakayama, central Japan. The ripened fruits were dried at a temperature of 40 °C for 6 h, 50 °C for 6 h, and then 60 °C for 2 h. The dried pericarps were then separated from the seeds using a sieve. HαS and HβS were obtained from MedChemExpress (Monmouth Junction, NJ, USA). Hydroxyl-ε-sanshool (HεS) was purchased from MuseChem (Fairfield, NJ, USA). Hydroxyl-γ-sanshool (HγS) was obtained from Chengdu Biopurify Phytochemicals Ltd. (Sichuan, China). Four types of surfactants with different Hydrophilic–Lipophilic Balance (HLB) values were used. They were polyglyceryl-2 oleate (Sunsoft Q-17B, HLB: 6.5), polyglyceryl-5 oleate (Sunsoft A-171E, HLB: 13.0), polyglyceryl-10 myristate (Sunsoft Q-14S, HLB: 14.5), and polyglyceryl-10 oleate (Sunsoft Q-17S, HLB: 12.0), all of which were obtained from Taiyo Kagaku Co. (Yokkaichi, Mie, Japan). MCTs (COCONARD MT^®^) were obtained from Kao Corporation (Tokyo, Japan). Rapeseed oil, high-performance liquid chromatography (HPLC)-grade acetonitrile, and α-Toc were purchased from Fujifilm Wako Chemicals (Osaka, Japan). Dextrin (Pineflow^®^) was obtained from Matsutani Chemical Industry Co., Ltd., Hyogo, Japan). All other chemicals were of analytical grade.

### 2.2. Extraction of Sanshools Using Various Solvents

Sansho pericarps were ground using an LM-PLUS laboratory mill (Osaka Chemical Co., Ltd., Osaka, Japan), and 0.8 g of the pericarp powder was suspended in 4 mL of various solvents. The suspension was stirred with a vortex mixer for 5 min and then filtered through Toyo filtering paper No. 1. The amount of HαS in the filtrate was measured using HPLC. Water, methanol, ethanol, 2-propanol, acetone, ethyl acetate, chloroform, n-hexane, rapeseed oil, and MCTs were used as solvents. The amount of HαS extracted by each solvent was indicated as a relative value by considering the HαS amount extracted by ethanol as 100%.

### 2.3. Preparation of Oil Extracts

Throughout the experiment, extraction protocols were developed by the authors. Sansho pericarps were ground using an LM-PLUS laboratory mill, and 8 g of pericarp powder was suspended in 40 mL of MCTs containing 0.1% α-Toc. The suspension was incubated at 50 °C for 15 min while stirring. After incubation, the suspension was filtered through Toyo filtering paper No. 1, and a filtrate was obtained. Next, 8 g of pericarp powder was added to the filtrate, and the stirring and filtration procedures were repeated. This extraction procedure was repeated one more time, and the resulting extract (MCT extracts) was used as the oil phase for emulsion preparation. Oil extracts without α-Toc were used as controls.

### 2.4. Preparation of Oil Powder

Oil powder (or powdered oil) is widely used in the food, pharmaceutical, and cosmetic industries. A powder was prepared by mixing the MCT extracts with an excipient. Sugar (lactose, maltose, or sucrose), dextrin, or starch were used as excipients. An excipient was added to the oil extract at a weight ratio of ≥3:1, and the mixture was stirred using a High-Flex homogenizer (AS ONE, Osaka, Japan) at 2000 rpm for 2 min.

### 2.5. Preparation of Emulsions

Emulsions were prepared using a modified method described by Koriyama [11]. First, 10 g of a surfactant and 104 g of glycerol warmed to 50 °C were transferred into a 500 mL stainless cup and mixed using a High-Flex homogenizer at 8000 rpm for 1 min. After 32 g of repeated-three-times MCT extracts with or without 0.32 g of α-Toc were added to the cup, all solutions were emulsified using the homogenizer at 8000 rpm for 5 min. Next, 48 mL of deionized water was added, and the entire mixture was further homogenized at 8000 rpm for 1 min. All emulsions were prepared by sonicating for 2 min, using 0.5 s pulses, at 70% amplitude using a UD-100 ultrasonic disrupter (TOMY DIGITAL BI-OLOGY Co., Ltd., Tokyo, Japan). Thus, a milky suspension was obtained. The final oil fraction of the emulsion was 16.5% (*w*/*w*).

### 2.6. Preparation of Powders with Emulsion (Powdered Emulsion)

A mixture of emulsions containing the powders was prepared by adding lactose to the emulsions at a weight ratio of ≥3:1 and stirring for 5 min at 2000 rpm using a High-Flex homogenizer. The obtained powder was subjected to accelerated testing, as described below.

### 2.7. Accelerated Testing of HαS-Stabilizing Activity

Samples were randomly placed in an incubator at a temperature determined for each experiment. After a fixed time of incubation, in the case of a liquid sample, 100 μL was taken from the sample and its weight was accurately measured. A fixed volume of the sample was removed and weighed. The samples were diluted with a fixed amount of isopropyl alcohol, and the amount of HαS was determined using HPLC. In the case of a powder sample, about 100 μg was collected from the sample and the weight was accurately measured. Next, 2 ml of deionized water was added to the sample and stirred for 1 min, and then 8 mL of isopropyl alcohol was added and stirred for 1 more min. The amount of HαS in the diluted solution was determined using HPLC. The accelerated test was repeated three times, and all samples were analyzed in triplicate. The HαS content of the accelerated test samples / the HαS content on day 0 × 100 [%] was calculated to obtain the residual rate (%).

### 2.8. HPLC Analysis for HαS

A Shimadzu LC-2010 instrument was used for the HPLC analysis. All samples were injected into an InertSustainSwift C18 4.6 × 250 mm column (GL Sciences, Tokyo, Japan). The column was eluted with a mixture of solvents A (30% acetonitrile) and B (80% acetonitrile) at a flow rate of 1.0 mL/min. The conditions were as follows: 0 min, solvent B = 0% → 35 min, solvent B = 45% → 50 min, solvent B = 100% → 51 min, solvent B = 0% → 55 min, solvent B = 0%. All analyses were conducted at 40 ℃ and monitored using UV absorption at 270 nm.

Several methods for the determination of sanshools based on reversed-phase HPLC analysis have been reported [12,13]. We modified the column, mobile phase, and gradient conditions of these methods [6]. The HαS of the standard samples listed in the Materials and Methods were diluted to 0.38, 0.76, 1.90, and 3.8 mmole/mL, respectively, using an ethanol solution containing 0.1% α-Toc and were analyzed using HPLC (*n* = 3) under the conditions described above. HαS was eluted with a single sharp peak around the retention time of 25.4 min. In this concentration range, a linear relationship was obtained between the HαS concentration and the HPLC peak areas. This calibration curve was used to determine the HαS content in a sample. The experiment was also performed three times on different days, but no daily difference was found. The LOD of this method was 0.00163 µmole/mL.

### 2.9. Measurement of Particle Size Distribution

A particle size analyzer (ELSZ 2000; OTSUKA ELECTRONICS Co., Ltd., Osaka, Japan) was used to measure the particle size of the emulsions.

### 2.10. Statistical Analysis

All data are expressed as the mean ± standard deviation (SD) values for each group. The statistical analysis was performed using a one-way analysis of variance with Student’s *t*-test. A value of *p* < 0.05 was considered statistically significant.

## 3. Results and Discussion

### 3.1. Extraction of Sanshools Using Various Solvents

Several solvents have been used to extract sanshools from the pericarps or leaves of the genus *Zanthoxylum*. Therefore, we investigated the extraction efficiency of various sanshool solvents. Solvents with different log P oct/wat values were selected, and the extraction efficiencies of these solvents were compared. The following solvents were used: water, methanol (−0.74), ethanol (−0.32), 2-propanol (0.8), acetone (−0.24), ethyl acetate (0.73), chloroform (1.97), *n*-hexane (3.9), rapeseed oil, and MCTs. The values in parentheses are the log P oct/wat values, which are a measure of lipophilicity or hydrophobicity. The amount of HαS in the ethanol extract was 25.0 ± 0.36 mg/mL (mean ± SD). Taking this amount as 100, the amount of HαS extracted with each solvent is shown as a relative value (Figure 1). All solvents except *n*-hexane showed high solubility for sanshools, and the amount of HαS extracted by chloroform was approximately 60% that of ethanol. Although chloroform has often been used as an extraction solvent for sanshools [1,4], this result does not imply that chloroform is an inferior extraction solvent. When chloroform was used as the extraction solvent, soaking and extraction were performed in a cool, dark place for 24 h [1,4]. However, for safety reasons, chloroform cannot be used in food. Since pepper oil is one of the traditional uses of huajiao, vegetable cooking oil could also be a useful solvent for extraction. Moreover, since the HPLC analysis did not show major differences in the types and levels of sanshools in these extracts, edible vegetable oil was considered one of the most suitable solvents for the application of sansho extracts in processed foods. Rapeseed oil and soybean oil were first used to extract sanshools. For example, the amount of HαS extracted by rapeseed oil was similar to that of MCTs, as shown in Figure 1. However, sansho rapeseed oil or soybean oil extracts began to smell like oxidized oil after one or two months at room temperature. When emulsified (described later) and left at room temperature for more than 12 months, the same oil oxidation smell was noticed. Therefore, we selected MCTs as an edible oil that does not contain unsaturated fatty acids and is not easily oxidized.

### 3.2. Stability of Sanshools in MCT Extracts

We previously reported that α-Toc has excellent HαS-stabilizing activity [6]. Therefore, MCTs with or without α-Toc were used to prepare the extracts. The amount of HαS in both extracts was 20.9 ± 0.26 mg/mL (mean ± SD). Both extracts were incubated at 50 °C and 70 °C, and the remaining amount of HαS during incubation was compared. As shown in Figure 2, the amount of HαS in both extracts was stable during 7 days of incubation at 50 °C. However, at 70 °C, the HαS content in MCT extracts with α-Toc was 85% of the original amount by day 7. On the other hand, the HαS content in MCT extracts without α-Toc became more unstable and significantly decreased to less than 20% after 7 days. These results indicated that αToc is useful for stabilizing HαS in MCTs.

Oil powder (powdered oil) is widely used in the food, pharmaceutical, and cosmetic industries. Therefore, we prepared a powder by mixing the MCT extracts with an excipient and examined the stability of HαS in the powder. Various excipients (sugar, dextrin, and starch) were examined, but there was little difference in the results. With sugar, we tried maltose and sucrose in addition to lactose. It became clear that none of the sugars were problematic for use in the powder. Lactose has no sweet taste, so we thought it would be appropriate for powdered oils. A powder with an MCT extracts/lactose ratio of 1:3 (*w*/*w*) was prepared, and this powdered oil is abbreviated as 1:3. If the ratio of lactose to emulsion is less than 1:3, it does not turn into a powder, but becomes mushy or sticky. We also prepared 1:10 and 1:21 powdered oil. As shown in Figure 3, the HαS content in the 1:3 containing α-Toc was stable at 50 °C for 7 days but gradually decreased and almost disappeared on the 14th day. At 70 °C, the HαS content in 1:3 containing α-Toc gradually decreased to approximately 70% or less on day 7 and disappeared on the 14th day. In contrast, in the 1:3 mixture containing no α-Toc, HαS disappeared within 7 days. Furthermore, increasing the ratio of lactose to MCT extract worsened the stability of HαS in the powders. For 1:10 and 1:21 powdered fats at 50 °C, α-Toc stabilized HαS, but at 70 °C the amount of HαS decreased to 64% after 1 week. Without α-Toc, little HαS was detected at 50 °C or 70 °C. This means that as the ratio of lactose to MCT extracts increases, the interaction between α-Toc and HαS rapidly disappears.

### 3.3. Stability of Sanshools in the o/w Emulsion

Simply mixing the MCT extracts of sansho pericarps with lactose resulted in a loss of stability of HαS. We hypothesized that the effect of α-Toc on the stability of sanshools was weakened in powdered oil because it was not possible to maintain the approach distance between the sanshools and α-Toc. One solution to this problem was to microencapsulate the extracts and maintain the sanshools and α-Toc in constant proximity. Glycerol, water, and various surfactants with different HLB values were added to the pericarp MCT extracts with or without α-Toc. The o/w emulsion was prepared as described in the Materials and Methods section. The amount of HαS in MCT extracts was 53.8 ± 1.8 mg/g, and the amount of HαS in this emulsion was 7.97 ± 0.24 mg/g. In addition, from the measurement of the particle size distribution meter, it was found that the diameter the of micelles in the resulting emulsions was 200–300 µm. By the way, the amount of these emulsions added to food is very small, and in some cases, the amount added is only 1/1000th of the amount of food. Therefore, the composition of the emulsion has little effect on the composition and properties of the food to which it is added. This is because sansho is a spice and is usually used in very small amounts.

The HαS content in each emulsion was monitored during an accelerated test at 50 °C. As shown in Figure 4, with or without the addition of α-Toc, the HαS in the emulsion decreased slightly at 50 °C for 7 days but was considered relatively stable.

As shown in Figure 3, the stability of HαS was not maintained in oil powder, even in the presence of α-Toc. Therefore, we powdered these emulsions, which we call “powdered emulsion” and investigated the stability of HαS in the powdered emulsions. An emulsion was prepared using Sunsoft Q-17B (HLB: 6.5) as a surfactant. This emulsion containing α-Toc was named the AT emulsion, and the emulsion without it was named the AN emulsion. The amount of HαS in these emulsions was 7.97 ± 0.24 mg/g. Lactose was added to each to make it into a powder. The AT and AN emulsions were mixed with lactose at ratios of 1:3 to 1:21, respectively. Each powder was incubated at 50 °C for 2 weeks to determine the remaining amount of HαS. As shown in Figure 5, in all powders of the AT emulsions containing α-Toc, the HαS content was maintained, even in the second week. This was very different from the case of the oil powder shown in Figure 3. On the other hand, for an AN emulsion that did not contain α-Toc, almost no HαS remained in the powder, even in the first week. This suggests that in the emulsion powder, the structure of the emulsion was maintained and the sanshools and α-Toc were always close. When other surfactants with different HLB values were used, results similar to those shown in Figure 5 were obtained. Through micellization, sanshools and α-Toc were always in close proximity and were isolated from the external environment by a membrane; therefore, it was thought that the oxidative decomposition of sanshool was suppressed.

Although the aroma components were not quantified, we also noticed that this emulsion retained the aroma of sansho for a long time. The future challenge is determining how much it can be maintained.

### 3.4. Effect of pH on the Stability of Sanshools

Sanshools are extremely unstable in acidic conditions [7]. This is thought to be due to the amide group in the chemical structure of sanshool. We dissolved an ethanol extract of sansho pericarps in buffer solutions of pH 2–12 and examined the stability of HαS, and HαS was found to be extremely unstable in the acidic buffer [6]. We attempted to investigate whether the stability of the sanshools in the emulsion was maintained when dispersed in an acidic solution.

An emulsion was prepared using Sunsoft A-171E (HLB: 13.0) as a surfactant. This emulsion containing α-Toc was named the BT emulsion, and the emulsion without it was named the BN emulsion. Each emulsion was mixed with the same amount of water or buffer solutions of pH 2.2 to 7 and incubated at 50 °C. The state of emulsification did not change significantly during the incubation time. As shown in Figure 6, between pH values of 3 and 7, the amount of HαS showed little decrease, even after 2 weeks; at pH 2.2, the amount of HαS decreased but to a lesser extent. This indicates that the sanshools in the BT emulsion were not affected by the pH. On the other hand, almost no HαS was detected in the BN emulsions in week one. In the emulsion of sansho pericarp MCT extract, the inside of the emulsion was isolated from the external environment by a surfactant film; therefore, it was thought to be unaffected by the pH of the liquid surrounding the emulsion.

Preventing oxidation and avoiding acidic conditions is an important requirement to stabilize sanshools [6,7]. Several antioxidants are useful for preventing oxidation [6]. The predominance of α-Toc among antioxidants may be related to the similarity of its chemical structure to that of sanshools. This may also be related to the high content of tocopherols in sansho pericarps [8]. However, it remains unclear how α-Toc approaches or binds to sanshools. Marquardt et al. investigated the positioning of α-Toc in a synthetic lipid bilayer using small-angle neutron diffraction, and their findings indicated that the phytol moiety penetrates the fatty-acid side-chain moiety and that the chroman skeleton is located in the hydrophilic portion of the phospholipid in the lipid bilayer [14]. Sanshool is also composed of hydrophilic and fat-soluble parts, and it is assumed that it occupies the same position as α-Toc in the lipid bilayer membrane.

Physical measures to prevent oxidation are also important. Vegetable oil microencapsulation has recently gained prominence in this regard [9,10,15]. Microencapsulation of functional substances (active food ingredients) dissolved in vegetable oils, such as aromatic components, unsaturated fatty acids, antioxidants, pigments, and vitamins, has been attempted [9,10,15]. Emulsion is one of the basic techniques for achieving microencapsulation and has long been used in cosmetics, pharmaceuticals, and foods. The o/w emulsion used in this study is the most basic emulsion and does not cost much. In this study, we found that when the spicy components of sansho coexist with α-Toc in an emulsion, the stability of HαS is extremely high. In addition, because this o/w emulsion prevents contact between the sanshools and acidic solvents to some extent, it may be possible to produce sansho-acidic beverages. Thus, limiting the micelle concentration is another important consideration. Tan et al. recently prepared HαS-loaded nanostructured lipid carriers (HAS-NLCs) and reported that in comparison with free HαS, HAS-NLCs showed 10.79-, 3.25-, and 2.09-fold higher stability against oxygen, light, and heat, respectively [16]. The use of α-Toc as a stabilizer for HAS-NLCs would have further enhanced its stability.

It has been reported that *N*-alkylamides widely occur in 25 plant families and have been used in foods [17]. Capsaicin, present in red pepper *(Capsicum annuum* L.), and piperine, present in black pepper (*Piper nigrum* L.), are also *N*-alkylamides [17]. These substances are responsible for the hot sensation of these peppers. Because these compounds are relatively stable, red and black pepper are produced all over the world and their use is enormous and diverse. If sanshool can be stabilized, it will lead to the expansion of its use in sansho as well. Sanshools have many biological activities and pharmacological effects, which include saliva-enhancing properties [13] and activation of the promotion of gastrointestinal motility [18]. Therefore, the stabilization techniques used in this study may be useful for the development of new products for sansho. Isobutylamides, useful components present in *Echinacea,* are also *N*-alkylamides and have been reported to be highly unstable substances. *Echinacea* is a popular herbal medicine in Western countries that has been used to treat colds and influenza [19,20]. However, its usefulness has been questioned in clinical trials [21]. Our findings may explain why clinical trials of *Echinacea* have been useful or ineffective. The stabilization techniques used in this study may be useful in solving these problems.

## 4. Conclusions

When sansho pericarps were treated with MCTs with α-Toc added, HαS was extracted. Glycerol, water, and surfactants were added to this extract and vigorously shaken to create an o/w emulsion. This emulsion was stably maintained, even when diluted with liquid or powder. We believe that this approach can stabilize the spiciness and aroma of sansho and will allow the development of various sansho-based foods and medicines.

## 5. Patents

TM, YY, and NY are named as inventors in patent applications (WO2021255943).

## Figures and Tables

**Figure 1 foods-12-03589-f001:**
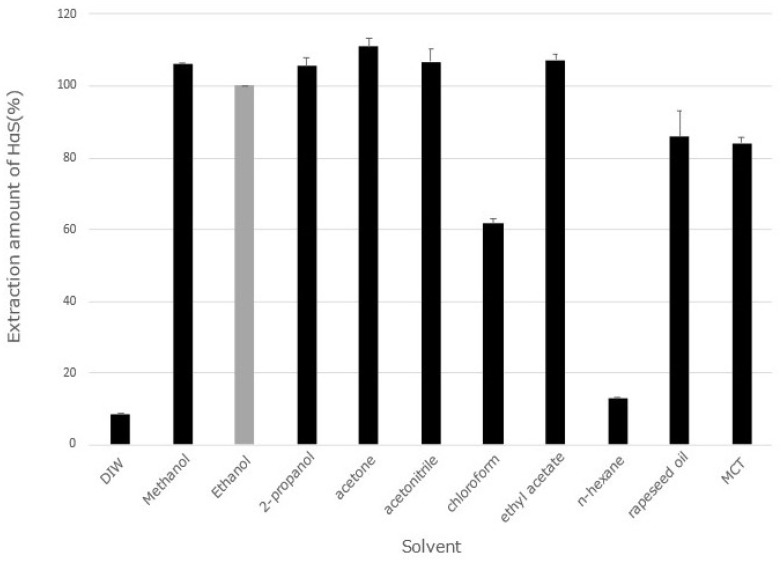
Amount of HαS extracted from sansho pericarps by various solvents. The amount of HαS extracted by each solvent is indicated relative to the amount of HαS extracted with ethanol (taken as 100). Data represent means ± SDs (*n* = 3).

**Figure 2 foods-12-03589-f002:**
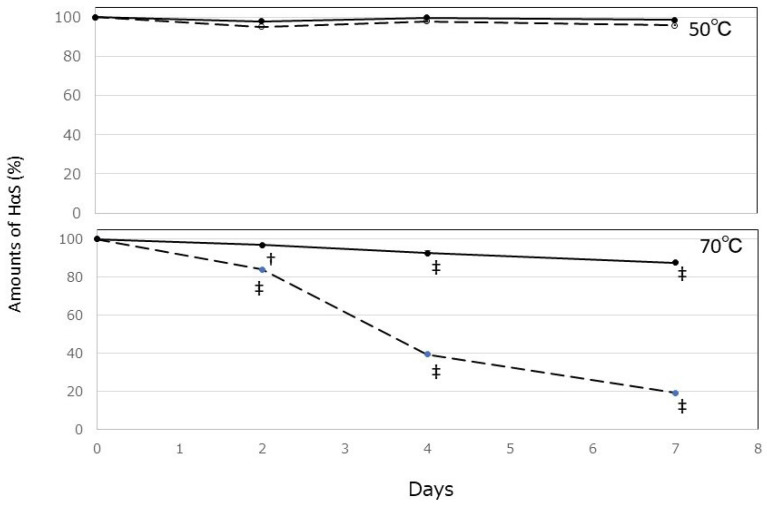
Effect of α-tocopherol on the stability of HαS in MCT extracts. The solid line indicates the MCT extracts with α-Toc, and the dashed line indicates the MCT extracts without α-Toc (no symbol: *p* > 0.05 vs. day 0; † *p* < 0.01 vs. day 0; ‡ *p* < 0.001 vs. day 0).

**Figure 3 foods-12-03589-f003:**
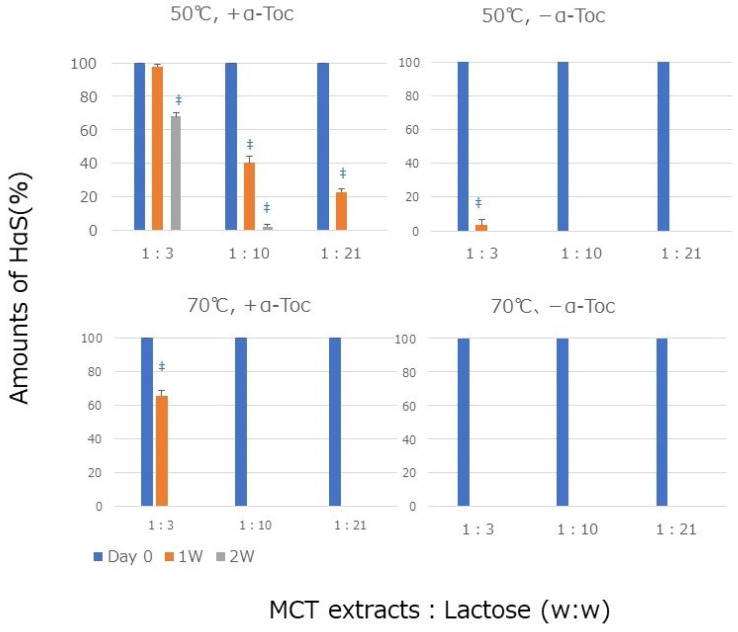
Stability of HαS in powdered oil. MCT extracts with or without α-Toc were mixed with lactose and incubated at 50 °C or 70 °C for 2 weeks. Data represent means ± SDs (*n* = 3) (no symbol: *p* > 0.05 vs. day 0; ‡ *p* < 0.001 vs. day 0).

**Figure 4 foods-12-03589-f004:**
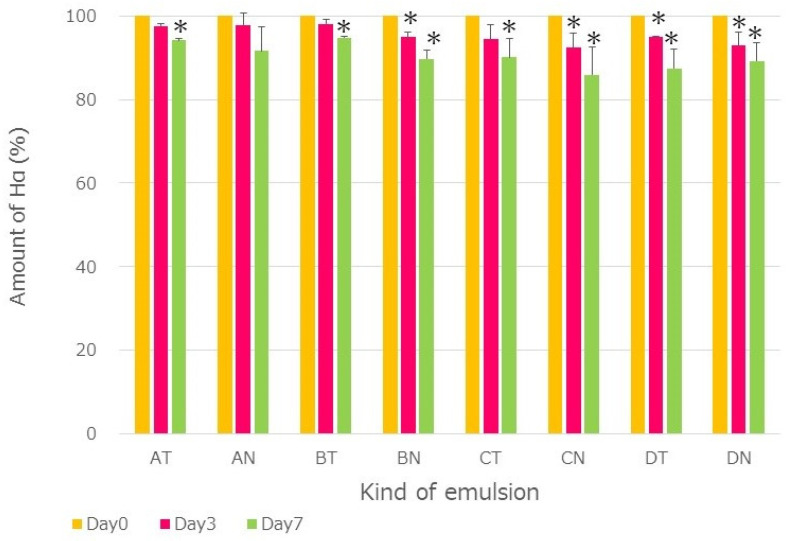
Stability of HαS in various o/w emulsions at 50 °C. A: Sunsoft Q-17B (HLB: 6.5); B: Sunsoft A-171E (HLB: 13.0); C: Sunsoft Q-14S (HLB: 14.5); D: Sunsoft Q-17S (HLB: 12.0); T: +α-Tocoferol; N: −α-Tocoferol (no symbol: *p* > 0.05 vs. day 0; * *p* < 0.05 vs. day 0).

**Figure 5 foods-12-03589-f005:**
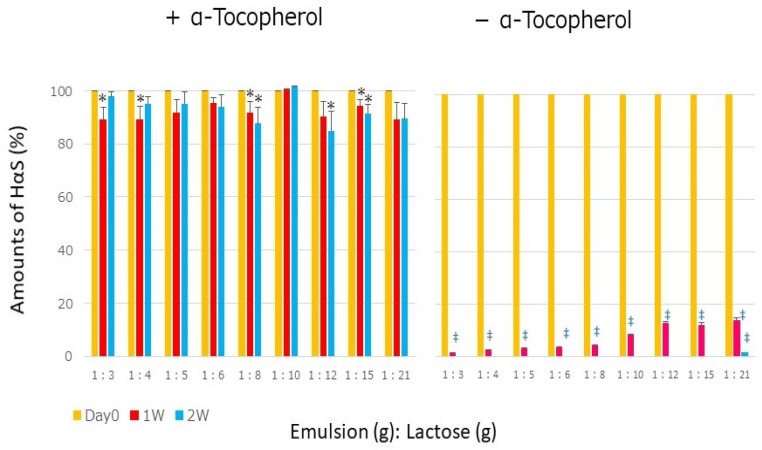
Stability of HαS in emulsion–lactose powder mixed at various ratios at 50 °C (no symbol: *p* > 0.05 vs. day 0; * *p* < 0.05 vs. day 0; ‡ *p* < 0.001 vs. day 0).

**Figure 6 foods-12-03589-f006:**
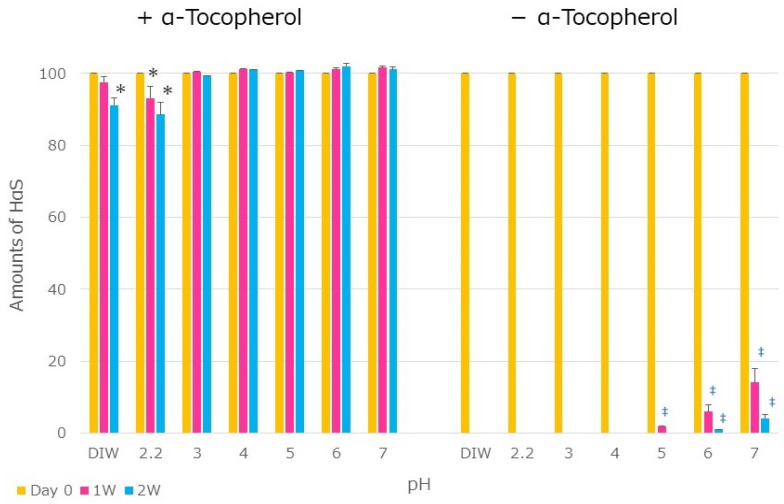
Stability of HαS in emulsions suspended in water or buffer (pH 2.2–7) at 50 °C (no symbol: *p* > 0.05 vs. day 0; * *p* < 0.05 vs. day 0; ‡ *p* < 0.001 vs. day 0).

## Data Availability

The data are contained within the article.

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
