# Peer review of "Stability of Hydroxy-α-Sanshool in Medium-Chain Triglyceride Oil and Corresponding Oil/Water Emulsions"

_foods, 2023, doi:10.3390/foods12193589_

Round 1

Reviewer 2 Report

The aim of this work was to determine the stability of extracts containing hydroxyl-α-sanshool, α-tocopherol and middle-chain triglycerides. Although the current paper may provide interesting results in the field of food science, the quality of the manuscript is not high enough to be published in the journal. There are a number of concerns with this manuscript such as clarity, and completeness of content.

Further comments:

-Introduction: it did not provide the properly background information for the tasks of this paper and motivation for this work is not well explained. The aim of the work is not provided.

-Material and Methods:

·        It is important to know which sample will be analysed in each case.

·       Why did you add lactose in the preparation of the emulsion? Why did you use ≥3:1 ratio?

·     Change the title of “HPLC” and “Determination of HαS”, You should explain the extraction, determination and quantification together.

·        Which surfactant did you use? Why did you use 104 g of glycerine? Can you report the proportion of oil in the emulsion? Why did you use this amount of α-tocopherol?

·        Did you prepare emulsions or nanoemulsions?

-Results and Discussion:

·        Justify better why authors used MCT as extraction solvent. “We selected MCT as the extraction solvent since it is not easily oxidized thereafter” Did you evaluate the other solvents?

-Conclusion: Provide the main finding of this work. Delete other information.

Reviewer 3 Report

Review of manuscript 2553411 for “foods” journal

The reviewer believes that this is a manuscript dealing with an interesting and actual topic of both technological and potentially commercial interest concerning the stability of hydroxy-α-sanshool in medium-chain triglyceride oil and corresponding oil/water emulsions. In preparing this scientific manuscript, the authors have considered the most relevant and updated literature evidence in this research field and presented their interesting results in a meaningful and comprehensive flow. A few comments /recommendations are given below for the consideration of the authors:

Introduction

- Please add a paragraph at the end of introduction section to elaborate on the innovative aspect of this manuscript/added value in this scientific field/? e.g. microencapsulation in sanshool was investigated for the first time?

Material Methods

(1) Section 2.2, 2.3, 2.5, 2.6 Extraction of sanshools by using various solvents, preparation of oil extracts/preparation of emulsions

Have you developed the extraction protocol at your laboratory? If yes, please explicitly mention this…otherwise please give literature references/sources for each used method..

(2) Section 2.9. HPLC Analysis/did you develop the method at your lab? Otherwise please provide a literature reference?

Discussion-Conclusions

- Lines 302-311….I don’t understand how the given info link to the results…For me most of presented info here would be more suitable for the introduction section…

- I would suggest some restructuring-reformulation.. maybe results & discussion sections could merge…and then have a better structured conclusion section where to present in a couple of bullet points what are the most important findings of the manuscript/any follow up work and market applications?

Reviewer 4 Report

Hi dear Editorial board and the respected authors

This article "Stability of hydroxy-α-sanshool in medium-chain triglyceride oil and corresponding oil/water emulsions” was revised and has a novelty and I recommend it after consideration of the following comments.

·         The type of statistical design used in this research should be mentioned.

·         Please include the results of your study as detail and data results must be utilize.

·         Please express the dependent and independent variables in the end of the introduction.

·         Please explain the novelty of the study as a comprehensively.

·         All Tables: The alphabetical statistical letters for the means should all be modified such that the greatest number has the letter a as the numbers go lower, letters b, c etc.

·         Fig 2. ‡P < 0.005 or ‡P < 0.001? Please recheck it again throughout the manuscript.

·         In Fig. 4 the D letter was omitted in the caption.

·         Fig. 5. It seems this data are close together. Please recheck themes.

·         Discussion text must grammar improve and in some cases it is very weak and maybe there is no discussion at all.

·         Conclusion is very general, try to make it more scientific, comprehensive and concise in detail, especially.

References: It is OK.

The article has many flaws in express and concept of English, it is suggested to be revised in a scientific and native way.

The article has many flaws in express and concept of English, it is suggested to be revised in a scientific and native way.
